# Investigating Both Mucosal Immunity and Microbiota in Response to Gut Enteritis in Yellowtail Kingfish

**DOI:** 10.3390/microorganisms8091267

**Published:** 2020-08-20

**Authors:** Thibault P. R. A. Legrand, James W. Wynne, Laura S. Weyrich, Andrew P. A. Oxley

**Affiliations:** 1Department of Ecology and Evolution, School of Biological Sciences, The University of Adelaide, Adelaide, SA 5005, Australia; laura.weyrich@gmail.com; 2CSIRO, Agriculture and Food, Hobart, TAS 7004, Australia; James.Wynne@csiro.au; 3South Australia Research and Development Institute, Aquatic Sciences Centre, West Beach, SA 5024, Australia; 4Department of Anthropology and the Huck Institutes of the Life Sciences, The Pennsylvania State University, State College, PA 16801, USA; 5School of Life and Environmental Sciences, Faculty of Sciences Engineering and Built Environment, Deakin University, Waurn Ponds, VIC 3216, Australia

**Keywords:** microbiota, immunity, fish, gut, skin, health, mucosa, aquaculture

## Abstract

The mucosal surfaces of fish play numerous roles including, but not limited to, protection against pathogens, nutrient digestion and absorption, excretion of nitrogenous wastes and osmotic regulation. During infection or disease, these surfaces act as the first line of defense, where the mucosal immune system interacts closely with the associated microbiota to maintain homeostasis. This study evaluated microbial changes across the gut and skin mucosal surfaces in yellowtail kingfish displaying signs of gut inflammation, as well as explored the host gene expression in these tissues in order to improve our understanding of the underlying mechanisms that contribute to the emergence of these conditions. For this, we obtained and analyzed 16S rDNA and transcriptomic (RNA-Seq) sequence data from the gut and skin mucosa of fish exhibiting different health states (i.e., healthy fish and fish at the early and late stages of enteritis). Both the gut and skin microbiota were perturbed by the disease. More specifically, the gastrointestinal microbiota of diseased fish was dominated by an uncultured *Mycoplasmataceae* sp., and fish at the early stage of the disease showed a significant loss of diversity in the skin. Using transcriptomics, we found that only a few genes were significantly differentially expressed in the gut. In contrast, gene expression in the skin differed widely between health states, in particular in the fish at the late stage of the disease. These changes were associated with several metabolic pathways that were differentially expressed and reflected a weakened host. Altogether, this study highlights the sensitivity of the skin mucosal surface in response to gut inflammation.

## 1. Introduction

The mucosal surfaces of fish, comprising the gut, skin, gills and olfactory organ act as the first lines of defense against pathogens and represent important primary barriers [1,2]. These surfaces are composed of various layers with different physical and chemical properties that protect the host from the environment and potential pathogens. More specifically, the mucosa are coated in a secretion of mucus, mainly composed of mucin, which acts as a physical barrier between the environment and the fish and limits the growth of microbes [3]. This mucus also houses an array of microbes called microbiota. Recent works on animals including fish have revealed that these microbial communities support important functions including the development and regulation of the immune response, and as such interact closely with the host immune system to fight infections and disease [2,4]. The mucosal surfaces also comprise lymphoid tissues (termed mucosa-associated lymphoid tissues: MALTs) which play an important role in the detection, recognition and defense against potential pathogens [2]. These MALTs contain cells responsible for both the innate and adaptive immune system of fish [5]. Thus, the interactions between these different systems are a fundamental feature in maintaining homeostasis and permeability within the mucosal surfaces.

The relevance of gut health in the farming of finfish has increased within recent years due to the emergence of various gastrointestinal disorders that have hindered the development of the industry [6]. Particular conditions such as enteritis (a gut inflammation) have become especially problematic in the farming of a number of different species such as Atlantic salmon (*Salmo salar*), zebrafish (*Danio rerio*), turbot (*Scophthalmus maximus*), yellowtail kingfish (*Seriola lalandi*), California yellowtail (*Seriola dorsalis*), pearl gentian grouper (*Epinephelus* sp.) and common carp (*Cyprinus carpio* L.) [7,8,9,10,11,12,13]. Dietary components including high supplementation of soybean meal in the feed seem to play an important role in the emergence of this disease (or other related inflammatory disorders) in several species [7,8,9,10,14,15]. However, it has been reported that this disease can also be induced by various pathogens, either bacterial (e.g., *Aeromonas hydrophila*) or parasitic (e.g., *Enteromyxum leei* and *Pseudocapillaria tomentosa*) [16,17,18]. Furthermore, studies on zebrafish have shown that the intestinal inflammation is dependent on the microbiota where specific microbiota can predispose an animal to this condition, highlighting the important role of microbial communities in the onset of inflammation [19]. In grass carp (*Ctenopharyngodon idellus*), it was shown that fish with inflammatory intestinal disorders had an altered gut microbiota associated with an increase in diversity in diseased fish [20]. In addition, in yellowtail kingfish suffering from enteritis, alterations in the microbiota of the skin and gills have also been observed, suggesting that broader, body-wide host responses may play a role in the dynamics of these communities [21].

Recent research has greatly enhanced our understanding of host–microbiota interactions in health and disease, in particular in mammalian systems [22]. It is now clear that there is a bidirectional relationship between the microbiota and the host, where the microbiota play an important role in the training and regulation of both the host’s innate and adaptive immunity which, in turn regulates and selects for specific bacterial assemblages across the different mucosal tissues [22]. As such, a balanced (homeostatic) state is thought to be required for normal functioning and defense against environmental stress (e.g., diet or antibiotics) [23]. Any dysregulation of this equilibrium is often linked to poorer performance and can ultimately result in disease [22]. However, in fish, host microbiota interactions are poorly understood, and research thus far has been mainly focused on gnotobiotic models such as zebrafish and threespine stickleback (*Gasterosteus aculeatus*) [24].

Studies investigating the fish immune response against diet induced gut inflammation have revealed useful biomarkers at the early stages of the disease including some proinflammatory cytokines and antioxidant enzyme related genes [11,13,25]. However, there is a lack of information regarding the interactions between the host and the microbiota during disease. Considering the importance of host–microbe interactions and the role of the fish microbiome in health and disease [26], further research is required to elucidate the underlying mechanisms leading to the development of the disease. Here, we investigated the influence of gut enteritis on the gut and skin microbiota of yellowtail kingfish and examined the host response using transcriptomics (RNA-Seq) to better understand the effect of a gut disease on the fish mucosal surfaces.

## 2. Materials and Methods

### 2.1. Experimental Design and Sample Collection

This study expands on earlier experimental work conducted on yellowtail kingfish that investigated the influence of gut enteritis on the outer surface (skin and gill) microbiota [21]. Specifically, in this study we expand on the earlier bacterial community (16S rDNA) analyses of the outer (skin) surfaces and surrounding environment (seawater) to include a comparison with the microbiota of the hindgut, as well as the assessment of the host response in the skin and hindgut tissues using transcriptomics (RNA-Seq). For this, RNA extracted from samples obtained by Legrand et al. [21] from a total of 36 fish of differing health states was used to generate Next Generation Sequencing (NGS) 16S rDNA amplicon libraries from the hindgut (for comparison with the data generated earlier for the skin (Accession number under the BioProject ID PRJNA396452)), and RNA-Seq libraries from the skin and hindgut. This included samples from 12 healthy fish (referred to herein as the “healthy” group) from a single seacage containing only individuals with no signs of infection, and 24 fish from a nearby seacage (<7 km) exhibiting signs of early and late stages of gut enteritis (herein referred to as the “early” and “late” groups, respectively). All fish were obtained and sampled under the auspices of a commercial aquaculture enterprise according to industry best practice veterinary care, with the health status of each treatment group confirmed by necropsy and histopathological assessment by farm health and veterinary personnel and an external pathology provider. All fish had been fed the same pelleted feed and came from the same hatchery run. For each fish, skin swabs were first collected upon netting by swabbing one side of the fish using FLOQSwabs^®^ (COPAN, Murrieta, CA, USA) and stabilized in tubes comprising RNAlater™ (Ambion, Austin, TX, USA). Fish were then euthanized using a lethal dose of AQUI-S (AQUI-S New Zealand Ltd., Lower Hutt, New Zealand) in seawater, and the gastrointestinal tract was carefully excised with a sterile scalpel. A scraping of the inner mucosal surface of the hindgut region was obtained using a sterile glass microscope slide. The contents of the hindgut were also stabilized in tubes comprising RNAlater™ (Ambion). All RNA samples were treated with the Turbo DNase free™ (Life Technologies, Carlsbad, CA, USA) kit to remove any residual gDNA, and were stored at −80 °C.

### 2.2. Library Preparation and Sequencing

In order to compare the bacterial (16S rDNA) community composition data obtained earlier from the skin, the same NGS 16S rDNA amplicon library preparation protocol was performed for the hindgut samples as conducted previously [21]. Briefly, purified total RNA extracted from the hindgut samples (as obtained using bead-beating and the RNeasy mini kit (Qiagen, Hilden, Germany) as detailed in Legrand et al. [21]) was converted into cDNA using the Superscript^TM^ III First Strand Synthesis System (Life Technologies, Carlsbad, CA, USA) according to the manufacturer’s instructions. The cDNA was subsequently concentrated by ethanol precipitation using standard procedures, quantified using the NanoDrop 2000 spectrophotometer (ThermoFisher Scientific, Indianapolis, IN, USA) and stored at −20 °C. The V1-V2 hypervariable region of the 16S rRNA gene was amplified from the cDNA from all hindgut samples (*n* = 36 fish) using a multistep approach using universal eubacterial primers 27F and 338R. Following the library preparation, samples were quantified and pooled in equimolar ratios before being sequenced on the MiSeq Illumina platform using 250nt paired-end sequencing chemistry through the Australian Genome Research Facility (AGRF, Melbourne, Australia). Raw demultiplexed sequencing data with sample annotations were deposited in the NCBI SRA data repository under the BioProject ID PRJNA637190.

For assessing host gene expression, transcriptomic (RNA-Seq) libraries were prepared from purified RNA extracts from a total of 9 skin and 9 hindgut samples (*n* = 3 per treatment group per sample type) following initial quality assessment using the LabChip System (Caliper Life Sciences, Inc., Hopkinton, MA, USA). Libraries were generated using the ScriptSeq^TM^ Complete Gold Kit (Epidemiology) (Illumina, San Diego, CA, USA), which included the initial depletion of the rRNA using the Ribo-Zero^TM^ Gold rRNA Removal Kit (Epidemiology) (Epicentre, Madison, WI, USA). For each library, a minimum of 100 ng of rRNA depleted RNA was used in each reaction according to the manufacturer’s instructions, and the libraries were purified using the MinElute^TM^ PCR Purification Kit (Qiagen). Potential contaminating primer dimers were removed by Exonuclease I treatment (Illumina) and further size selection of fragments (~200–600 bp) using the SPRIselect Reagent Kit (Beckman Coulter, Brea, CA, USA). Fragments were then assessed for quantity and quality using the Quant-iT™ picogreen™ dsDNA Assay Kit (Invitrogen) and the LabChip System (Caliper Life Sciences, Inc.). Libraries were pooled in equimolar ratios, and 6 samples were multiplexed per lane and sequenced on the Illumina HiSeq4000 platform (Illumina) using 150nt paired-end sequencing chemistry through the Murdoch Children’s Research Institute (MCRI)—Translational Genomics Unit (Melbourne, Australia). Raw demultiplexed sequencing data with sample annotations were deposited in the NCBI SRA data repository under the BioProject ID PRJNA639544.

### 2.3. Bioinformatics and Statistical Analysis

The raw 16S rDNA sequence reads obtained from 33 hindgut samples (3 samples failed in library preparation which were fish #114, #128 and #129) were paired using PEAR (v. 0.9.5), and the primer regions were removed [27]. These trimmed sequence reads were subsequently merged with the 36 skin and 2 seawater sample fastq files obtained in the earlier study of Legrand et al. [21] (NCBI SRA accession numbers under the project PRJNA396452) and were processed and analyzed together using the QIIME2 (v. 2019.1) pipeline [28]. Demultiplexed paired-end sequence reads were truncated to a length of 320 bp, quality filtered and denoised into amplicon sequence variants (ASVs) using the DADA2 plugin [29]. A total of 3,183,303 demultiplexed paired-end sequence reads were assigned to 9155 ASVs features from a total of 71 samples. The number of reads ranged from 16,255 to 135,287 with a median of 40,799 per sample. Following the denoising and removal of reads associated with chloroplast, mitochondria and eukaryotes (after assigning taxonomy), a total of 3,116,835 reads were obtained for downstream analysis. Each sample was rarefied to a depth of 16,255 reads, resulting in a total of 7863 ASVs in the dataset. Alpha rarefaction showed sufficient coverage of the samples (Appendix A). Taxonomy was assigned to the ASVs using the q2-feature-classifier against the Silva 132 99% OTUs (Operational Taxonomic Units) reference sequences resource [27]. Alpha-diversity metrics (Shannon’s diversity, Pielou’s evenness and Chao1 richness), beta diversity metrics (Bray–Curtis), and Principle Coordinate Analysis (PCoA) using both the weighted and unweighted Unifrac distance matrix were estimated using q2-diversity. Statistical differences for alpha diversity were assessed using the Kruskall–Wallis test with Benjamini-Hochberg correction for pairwise comparison. Statistical differences for the ASV dataset as a whole were identified using PERMANOVA. QIIME artifacts were imported into R using the package Qiime2R, and plots were made using Phyloseq and ggplot2 [30]. Statistical differences for each ASV (differential abundance) were assessed using Deseq2 [29], as suggested recently for the analysis of microbiome data with a small number of replicates per treatment (<20) [31,32].

Sequencing of the transcriptomic libraries from the 9 hindgut and 9 skin samples yielded ~1064 million reads, with an average of 59 ± 17 million reads per sample (Appendix A). Reads were quality filtered to remove low quality reads and Illumina adapters using Trimmomatic (v0.38) with the parameters ILLUMINACLIP:TruSeq3-PE-2.fa:2:30:10 LEADING:10 TRAILING:10 SLIDINGWINDOW:4:20 MINLEN:40 [33]. Then, rRNA sequence reads were removed from the dataset using SortmeRNA (v. 2.1) by using the default settings, which included interrogation against the SILVA rRNA database [34]. A total of 476 million cleaned paired-end reads (average of 26 ± 13 million reads per sample) were subsequently obtained and, in the absence of an annotated reference genome for *S. lalandi*, were mapped to the genome from the related species *Seriola dumerili* (accession number GCA_002260705.1 in ensembl.org) using STAR (v. 2.5.3a) [35]. Reads were aligned back to the genome and counted with Subread (v. 1.6.2) using the function featureCounts [36]. Approximately 80 ± 2% of the reads was able to be mapped to the *S. dumerili* genome (a similar mapping rate around ~80% was found using the *S. lalandi* reference genome). The resultant count data were used to initially identify biological outliers via ordination of the relative abundances of the transcript sequence reads using Deseq2 (wherein one hindgut sample from the early group was removed from the downstream analysis (Appendix A)), and Deseq2 was used to calculate differential gene expression [29]. Genes were identified as significantly differentially expressed when p-adj < 0.05 and log2 fold-change> or <1.5. Differential pathway analysis was performed using Voronto [37] and the R package clusterProfiler [38] using the KEGG (Kyoto Encyclopedia of Genes and Genomes) database [39].

## 3. Results

### 3.1. Analysis of the Gut and Skin Microbiota

To explore whether the impact of gut enteritis on bacterial community dynamics is similarly reflected across the mucosal surfaces of yellowtail kingfish, hindgut samples from fish belonging to three different health states (*n* = 12 fish per health state) were compared with data obtained earlier from the skin of the same fish and environmental (seawater) samples [21].

The overall gut microbiota was significantly different from the skin samples based on the Bray–Curtis similarity matrix (Pseudo-F = 38.80, *p* = 0.002). In addition, the gut microbiota had a significantly lower Shannon diversity than the skin (*p* < 0.001) and significantly lower ASV richness (*p* < 0.001) with 40 ± 20 ASVs when skin samples had an average of 545 ± 296 ASVs per sample.

#### 3.1.1. Influence of Gut Enteritis on the Global Gastrointestinal and Skin Mucosal Microbiota

We investigated the effect of gut enteritis on the gastrointestinal and skin mucosal bacterial communities. We found that the health status had a significant influence on the global gut (Figure 1a; Pseudo-F = 4.43, *p* = 0.003) and skin (Figure 1b; Pseudo-F = 4.55, *p* = 0.003) bacterial communities based on the weighted Unifrac distance matrix. More specifically, the gut microbiota of healthy fish were significantly different when compared to fish at the early (Pseudo-F = 7.05, *p* = 0.045) and late stage of the disease (Pseudo-F = 4.99, *p* = 0.045). However, the gut microbiota of the fish at the early and late stage of the disease were not significantly different (Pseudo-F = 0.44, *p* = 0.615). In contrast, no global bacterial communities differences were found when using the unweighted Unifrac distance (Appendix A, Pseudo-F = 1.15, *p* = 0.268). On the other hand, the skin microbiota were significantly different between all health states using both weighted Unifrac (*p* < 0.05 for all pairwise comparisons) and unweighted Unifrac distances (Appendix A, *p* < 0.001 for all pairwise comparisons).

The alpha-diversity (Shannon index) in the gut microbiota of the fish at the late stage of the disease was lower than both healthy and early fish, though not significantly different (Figure 1c; *p* = 0.123 and *p* = 0.291, respectively). Similarly, the microbial community evenness (Pielou’s index) was lower in the fish at the late stage of the disease but was not significant, while the richness (Chao1 index) was fairly consistent across all three health status (Appendix A). Unlike the gut, fish at the early stage of the disease exhibited a significant loss of Shannon’s diversity in the skin compared to both healthy and fish at the late stage of the disease (Figure 1d; *p* = 0.007 and *p* = 0.020, respectively). This was associated with a significant loss of both evenness and richness in the fish at the early stage of the disease when compared to both healthy and early stages of the disease (Appendix A).

#### 3.1.2. Taxonomic Composition and Potential Biomarkers of Gut Enteritis in the Gut and Skin Microbiota

The gut microbiota was dominated by a few bacterial members, including *Mycoplasmataceae*, Aliivibrio, Photobacterium and Brevinema (Figure 2a). The most dominant ASV was associated with an uncultured *Mycoplasmataceae* sp. and represented 54 ± 34% of the total relative abundance in the gut samples. This ASV was significantly less prevalent in healthy fish than both fish at the early and late stage of the disease (Figure 2b; *p* = 0.021 and *p* = 0.013, respectively, using the Kruskall–Wallis test). At the genus level, the other most dominant members were Photobacterium (13% of the total relative abundance), Aliivibrio (11%), Brevinema (10%) and Vibrio (9%). In contrast, the skin microbiota was more diverse and dominated by other bacterial lineages in both health states. At the order level, the skin microbiota was dominated by Flavobacteriales members, representing 46% of the total relative abundance (Figure 2c). Other important members were related to Alteromonodales (10%), *Rhodobacterales* (9%), Oceanospirillales (5%) and Synechococcales (3%).

To further characterize the change of gut and skin microbial communities associated with the disease, we performed some differential abundance analyses to identify potential biomarkers within these mucosal tissues. In the gut, we found 12 ASVs that were significantly differentially abundant between healthy fish and fish at the early stage of the disease (Appendix A). All of them were less abundant in the fish at the early stage, with four associated with Aliivibrio and four associated with Photobacterium. Within the fish at the late stage of the disease, we found 11 ASVs that were differentially abundant when compared with healthy fish (all less abundant in diseased fish) (Appendix A). Among these, we found the same ASVs related to Aliivibrio and Photobacterium, indicating that these ASVs are less abundant in diseased fish, regardless of the stage of the disease. Finally, only one ASV was significantly different between fish at the early and late stage of the disease, which was Gammaproteobacteria (Appendix A).

In stark contrast to the gut, we found more differentially abundant ASVs associated with disease within the skin. Interestingly, while most of the differentially abundant ASVs in the gut were found in the two diseased states (both early and late), those found in the skin were mainly in the early condition only. More specifically, 195 and 263 ASVs were found differentially abundant between fish at the early stage of the disease and healthy and late condition, respectively (Appendix A). However, despite being housed in two different cages with relatively distinct water bacterial communities (Figure 2c), we only found 18 differentially abundant ASVs between healthy fish and fish at the late stage of the disease (Appendix A). At the genus level, Alteromonas, Pseudoalteromonas, Glaciecola, Halomonas, Marinobacter, Cobetia, Idiomarina, Arcobacter, SAR 92 clade, Synechococcus CC9902 and Litoricola were all found to be significantly depleted in the fish at the early stage of the disease when compared to healthy fish or fish at the late stage of the disease.

### 3.2. Analysis of the Transcriptomics Data

In order to evaluate the mucosal immunity across both the gut and skin mucosal surfaces of yellowtail kingfish in response to gut enteritis, we performed some differential gene expressions on selected samples representing the three different conditions (i.e., healthy, early and late stage of enteritis; *n* = 3 per condition). We found <100 significantly differentially expressed genes (DEGs) in the gut when comparing all treatment groups (Table 1). On the other hand, more DEGs were found in the skin, in particular in the group at the late stage of the disease (1467 DEGs when compared to healthy and 2068 DEGs when compared to early).

#### 3.2.1. Differential Expression in the Gut of Fish Exhibiting Different Health States

Of particular interest, we found a number of genes associated with the intestinal immune network for immunoglobulin production (H-2 class II histocompatibility antigen and HLA class II histocompatibility antigen associated genes) and the Toll-like receptor signaling pathway (TIR domain containing adaptor protein and Toll-like receptor 2 type-2) downregulated in the fish at the late stage of the disease (Appendix A). In addition, we found a neutrophil related gene (*ncf4*) downregulated in these late fish (log2fold = −1.58); this gene group plays a role in the phagosome pathway and a procathepsin H-like gene involved in the lysosome and apoptosis pathway. In contrast, some genes related to glycerolipid metabolism (patatin-like phospholipase) and glycine metabolism (glycine dehydrogenase and glycine decarboxylase) were found upregulated in the fish at the late stage of the disease (Appendix A).

In the fish at the early stage of the disease, we found an upregulation of the histone H2A-like gene involved in the necroptosis pathway (log2fold = 6.15, Appendix A). On the other hand, two genes involved in the apoptosis pathway (procathepsin H-like and inositol 1,4,5-trisphosphate receptor type 1-like) were downregulated in these early group fish (Appendix A). Overall, due to the low number of DEGs found in the gut, no pathways were significantly differently expressed between groups.

#### 3.2.2. Differential Gene Expression and Associated Pathways in the Skin of Fish at the Late Stage of the Disease

Due to the high number of DEGs found in the skin, we used Voronoi tessellation diagrams (using Voronto) to represent the expression of the different pathways between the different health states. In the fish at the late stage of the disease, the immune system pathway was downregulated compared to healthy fish (Figure 3). In fact, all six pathways at level 3 of the KEGG database (including but not limited to the Toll-like receptor signaling pathway, NOD-like receptor signaling pathway and intestinal immune network for immunoglobulin production pathway) were downregulated. In addition, we found a downregulation of the cytokine–cytokine receptor interaction pathway including the downregulation of cytokines (such as *il1*, *il8*, *il12*, *il17* and *il23*) and chemokines (Figure 3, Appendix A). In contrast, the ECM (extracellular matrix)–receptor interaction and focal adhesion pathways were upregulated in these fish at the late stage of the disease (Figure 3).

We then investigated the significantly differentially expressed pathways in these fish at the late stage of the disease. Using clusterProfiler, we identified six significantly downregulated and four significantly upregulated pathways (Figure 4). The most significant pathways were ECM–receptor interaction and focal adhesion (both upregulated) and the cytokine–cytokine receptor interaction pathway (downregulated). The upregulation of the ECM-receptor interaction was characterized by an upregulation of collagen, laminin, reelin, thrombospondin, fibronectin and tenascin (Appendix A).

#### 3.2.3. Differential Gene Expression and Associated Pathways in the Skin of Fish at the early Stage of the Disease

Although no significant differentially expressed pathways were identified, numerous genes were differentially expressed in the fish at the early stage of the disease compared to healthy fish (Appendix A). In contrast to the fish at the late stage of the disease, we found that the cytokine–cytokine receptor interaction pathway was upregulated in the early stage fish compared to healthy fish (Figure 5). The immune system (including four out of five pathways at level 3) was also upregulated in the fish at the early stage of the disease. Furthermore, we found a notable upregulation of the foxO signaling pathway due to the strong upregulation of the recombination-activating gene (*rag1*, log2fold = 6.74). The cellular community pathways were downregulated (including all four associated pathways at level 3 comprising focal adhesion, tight junction, gap junction and adherens junction) in these early fish.

## 4. Discussion

To date, enteritis remains a major issue in the farming of numerous carnivorous species fed a diet partly constituted of soybean meal. Numerous strategies have been deployed to mitigate soybean-induced inflammation in fish. These include the supplementation of glutamine, arginine, resveratrol, microalgae, bacteria grown on natural gas and lactoferrin in the feed [40,41,42,43,44]. Fermentation of soybean meal prior to feeding has also been tested in turbot with encouraging results, suppressing the intestinal inflammation and enhancing the intestinal integrity [45]. Furthermore, efforts in selecting resistant fish with increased tolerance to plant diets have been made in some species such as rainbow trout (*Oncorhynchus mykiss*) [46]. In yellowtail kingfish, efforts in finding alternatives to soybean meal showed that poultry byproduct, faba bean and lupin kernel meals represent good protein sources compared to corn gluten and blood meals [47]. Here, we investigate both the microbiota and gene expression across the gut and skin mucosal surfaces of healthy fish and fish at different stages of enteritis, identifying important microbial changes and differentially expressed host genes and associated pathways that reflect the health status of the fish.

The gastrointestinal microbiota of diseased farmed yellowtail kingfish was dominated by an uncultured *Mycoplasmataceae* sp. Members of this family are typically found in the gut of farmed fish including Atlantic salmon, rainbow trout and common carp [48,49,50]. In our study, we found that this member was prevalent in diseased fish (both at the early and late stage of gut enteritis) when compared to healthy fish, indicating a potential harmful effect of these bacteria. In zebrafish, Mycoplasma was prevalent in fish exposed to a parasite (*Pseudocapillaria tomentosa*) and positively correlated with hyperplesia [18]. It was hypothesized that this member was responsible for the lesions in the fish, confirming a potential harmful role of certain *Mycoplasma* sp. In fact, this genus already includes known fish pathogens, such as *Mycoplasma mobile*, a bacterium colonizing the gills of freshwater fish [51]. In contrast, recent genome reconstruction analysis revealed a mutualistic lifestyle of new *Mycoplasma* species isolated from Atlantic salmon and hadal snailfish (*Pseudoliparis swirei*) [50,52]. In our study, it is unclear whether this bacterium played a role in the disease, or was the result of the poor overall health of the fish. Thus, functional analyses (e.g., shotgun metagenomics or metatranscriptomics) may be further required to elucidate the role of this bacterium in yellowtail kingfish health.

The overall gut bacterial community was significantly different between healthy and diseased fish, and was associated with a loss of diversity in the fish at the late stage of the disease. This is supported by the literature, where numerous investigations reported a loss of microbial diversity in fish exposed to stress or disease [26]. In addition, the overall skin microbiota was significantly different between all three health states, indicating that this gut disease not only influences the gut microbiota but also the outer surface microbiota, as previously shown by our group [21]. We found the largest number of differentially abundant ASVs between fish at the early stage of the disease and fish at the late stage of the disease, even though these fish were housed in the same cage. In contrast, very few differentially abundant ASVs were detected between healthy and fish at the late stage of the disease despite them being housed in two different cages separated by almost 7 km. While the fish skin microbiota has been shown to be a lot more sensitive to the environment than the gut [53], our results shows that health status can also play a major role in shaping the fish skin microbiota regardless of the surrounding environment. Interestingly and in contrast to the gut microbiota, we found a drastic loss of skin microbial diversity and evenness in the fish at the early stage of the disease when compared to healthy fish and fish at the late stage of the disease. Loss of diversity and evenness are characterized by a reduced resilience and functional capacity of the microbial communities, indicating that the gut and skin microbiota of diseased fish may have lost important functions including resistance to opportunistic pathogens [54].

To better understand this loss of bacterial diversity in the fish skin at the early stage of the disease, we investigated the immune response through transcriptomic (RNA-Seq) analysis. Firstly, we found that the immune system pathway was upregulated in the fish at the early stage. More specifically, the cytokine–cytokine receptor interaction pathway was upregulated in this condition compared to healthy fish, and significantly upregulated when compared to fish at the late stage of the disease. In this pathway, interleukin 8-like, regakine 1-like and CXC (Cystine-X-Cystine, where X is any amino acid) chemokine receptor type2-like genes were significantly upregulated in the fish at the early stage of the disease compared to the other two health states. These genes could potentially be used as biomarkers for the early detection of gut enteritis in yellowtail kingfish, although this requires further validation. Interleukin 8 was found to modulate the early cytokine immune response in rainbow trout, and its upregulation at the early stage of the disease indicates a proinflammatory response in the fish skin [55]. This same gene was found to be upregulated in the skin of rainbow trout after Ich infection, suggesting that its expression could be a potential useful biomarker for the detection of a proinflammatory response in fish [56]. CXC motif chemokine receptor 2 (*cxcr2*) upregulation has been associated with numerous inflammations and is known to induce the recruitment of neutrophils, supporting the idea of an immune response at this stage of the disease [57]. In addition, we found a strong upregulation of the foxO signaling pathway in the early stage fish, driven by a significant upregulation of recombination activating gene 1 (*rag1*). The protein encoded by this gene is involved in antibody and T-cell receptors and as such, plays an important role in the recognition of pathogen and immunoregulation. In fact, *rag1* is a known gene marker for the early development of the fish immune system, and therefore, the upregulation of this gene at the earliest stage of the disease confirmed an immune response in the skin mucosal surface [58]. In support of this, a recent study showed that *rag1*-deficient zebrafish did not develop intestinal inflammation when fed an inflammatory diet constituted of soybean, in contrast to normal fish [11]. This highlights the role of adaptive immunity in the response against enteritis, a feature confirmed in yellowtail kingfish. Altogether, these changes in gene expression suggest a strong immune response located in the fish skin. Some changes also highlight compromised functions within this mucosal surface. For instance, we observed a downregulation of the adherens junction, gap junction and tight junction—three pathways extremely important for barrier function [59]. Considering the vital role of the skin as a physical barrier to prevent the intrusion of potential pathogens, these changes could lead to the dysfunction of this organ, resulting in an increased disease susceptibility [60].

In the skin of fish at the late stage of the disease, we observed a significant upregulation of the ECM–receptor interaction and focal adhesion pathways. The ECM–receptor interaction pathway plays important functions in the host immune system including cell proliferation, differentiation and survival, intercellular communication and the regulation of leukocytes into inflamed tissues [61]. The upregulation of several proteins involved in this pathway likely reflect the advanced stage of the disease in which the host is developing a strong response to repair tissues. The upregulation of the focal adhesion pathway is not surprising considering its close interaction with the ECM–receptor interaction pathway. Indeed, focal adhesion also plays a role in the regulation of cell cycle progression and its dysregulation has been shown in numerous human diseases such as cancer and Alzheimer’s disease [62,63]. Within this pathway, we found an upregulation of integrin alpha subunit (ITGA) associated genes. These integrins are also involved in the ECM pathway and can be characterized as signaling molecules controlling cell differentiation, growth and survival [64]. In contrast to the fish at the early stage of the disease, fish at the late stage showed a downregulation of numerous cytokines (e.g., chemokines and interleukins) in the skin. This would suggest that the fish have passed the acute immune response phase and is reflective of a very weakened host. The downregulation of the immune system across the six KEGG pathways at level 3 confirms this hypothesis.

Although we investigated the influence of a gut disease, few genes were differentially regulated during disease in the gastrointestinal tract. Recently, it was shown that grass carp can exert different responses along the intestinal tract to induce inflammation when fed soybean ß-conglycinin [65]. More specifically, while the midgut and hindgut showed signs of inflammation, no changes were found in the foregut. Similarly, it was found that rainbow trout exhibits different microbial and immune responses across different regions of the digestive tract (e.g., mouth, pharynx, stomach, foregut, midgut and hindgut) following viral infection [66]. In this study, we only investigated the hindgut gene expression and microbiota, and therefore, other parts of the intestinal system could have had differential gene expression and/or microbial changes that went undetected.

Recently, the concept of the hologenome has gained popularity. This concept argues that the genome of the host and its associated microbial communities (microbiome) are in constant interaction, and as such, cannot be viewed independently [67]. It is well known that fish regulate their microbiota across all mucosal surfaces through mucus production, immune related cells and antimicrobial peptides [2]. Here, we speculate that the loss of microbial diversity observed in the skin of fish at the early stage of the disease could be linked with the upregulation of the immune system. More specifically, the upregulation of *rag1*, *il8* and *cxcr2* may be in part responsible for the observed changes in the skin microbiota. Moreover, *cxcr2* (also called *il8rb*) encodes a protein which is a receptor for *il8*. As such, *il8* and *cxcr2* interact closely and the upregulation of them both is not surprising. Since *il8* is associated with antimicrobial responses [66,68,69,70], its upregulation in the skin of fish at the early stage of the disease may have played a role in this observed loss of microbial diversity. Furthermore, the microbiota is known to modulate the host immune response [71]. More specifically, the production of several inflammatory cytokines can be driven by specific microbiota, highlighting the importance of host–microbiota interactions [72]. In this study, the number of replicates was insufficient in order to correlate the host gene expression with microbial diversity and further work should be performed in order to better understand host–microbe interactions, in particular in a health and disease context.

## 5. Conclusions

In this study, we found that gut enteritis perturbed the yellowtail kingfish gut microbiota, with an enrichment of an uncultured *Mycoplasmataceae* sp. in diseased fish. We observed profound changes within the skin microbiota, highlighting the sensitivity of this mucosal surface in relation to the host health. More specifically, fish at the early stage of the disease had a significant loss of microbial skin diversity when compared to both healthy fish and fish at the early stage of the disease. Surprisingly, gene expression within the gut did not widely differ between health conditions. In contrast, numerous differentially expressed pathways and genes were found in the skin, particularly in the fish at the late stage of the disease where several metabolic pathways were differentially expressed.

## Figures and Tables

**Figure 1 microorganisms-08-01267-f001:**
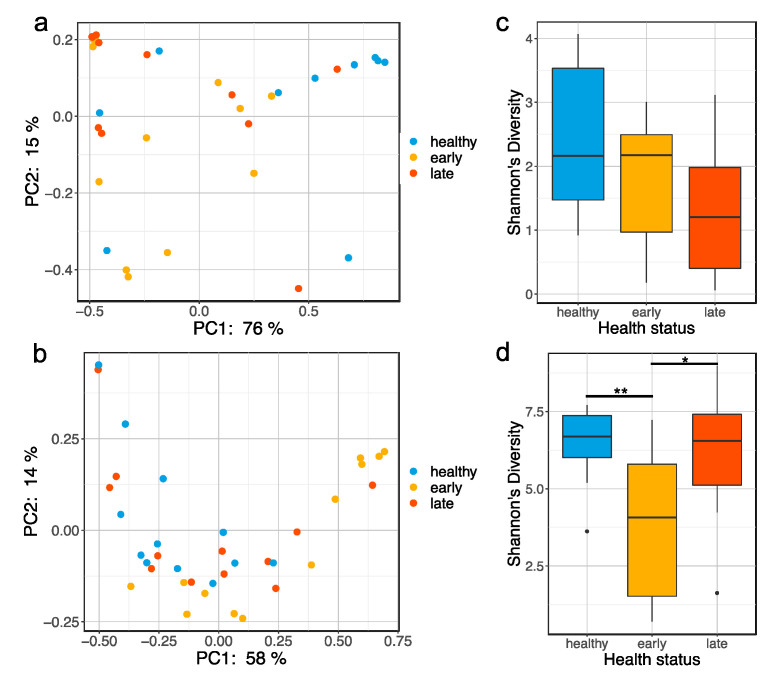
Global bacterial community changes associated with gut enteritis. PCoA plot based on the weighted Unifrac distance matrix showing clustering of gut (**a**) and skin (**b**) microbiota samples by health status (e.g., healthy, early stage of enteritis and late stage of enteritis); boxplot representing the Shannon’s diversity of the gut (**c**) and skin (**d**) microbiota for the different health status. Statistical differences were assessed using a Kruskall–Wallis test, with the levels of statistical significance between groups denoted by asterisks, with alpha set at 0.05.

**Figure 2 microorganisms-08-01267-f002:**
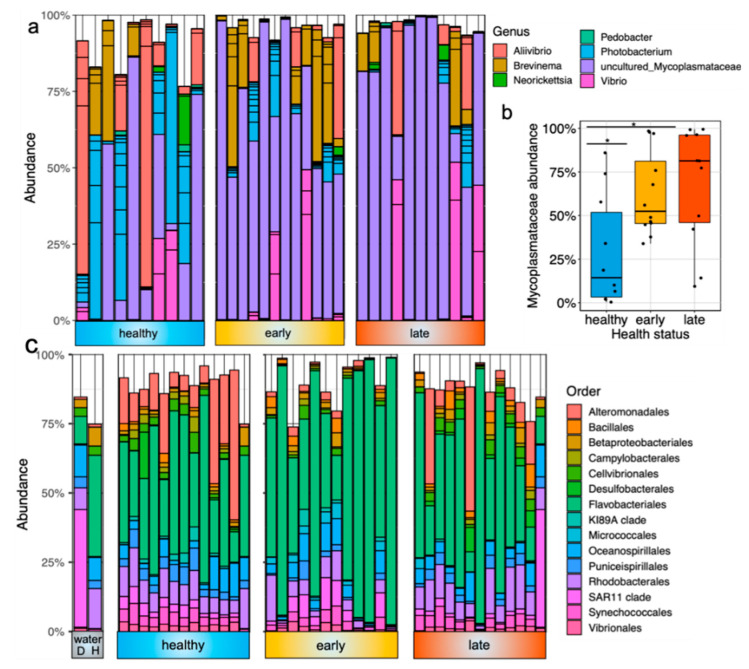
Taxonomic composition of the fish gut and skin microbiota with different health states: (**a**) barplot representing the relative abundance of the top 30 most abundant ASVs in the gut microbiota of fish exhibiting different health states (e.g., healthy, early stage of enteritis and late stage of enteritis); (**b**) boxplot representing the relative abundance of an uncultured *Mycoplasmataceae* sp. in the gut microbiota of fish exhibiting different health states; statistical differences were assessed using Wilcoxon test, with the levels of statistical significance between groups denoted by asterisks, with alpha set at 0.05; (**c**) barplot representing the relative abundance of the top 15 most abundant order in the skin microbiota of fish exhibiting different health status as well as seawater bacterial communities (D = diseased cage and H = healthy cage).

**Figure 3 microorganisms-08-01267-f003:**
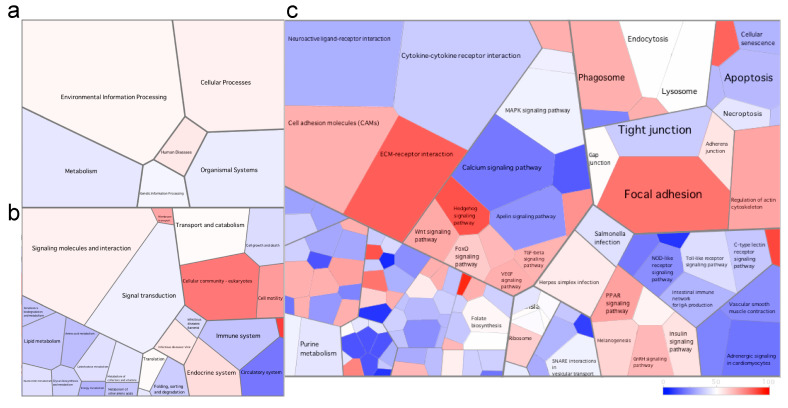
Voronoi tessellation diagrams representing differentially expressed pathways at (**a**) level 1, (**b**) level 2 and (**c**) level 3 of the KEGG database in the skin of the fish at the late stage of the disease. Each polygon represents an ontology term, with their size corresponding to the numbers of genes involved in the associated pathway. To further explore the pathways differentially expressed between health states, we imported all DEGs (and associated logs values) to generate the Voronoi tessellations. Pathways colored in red are upregulated, and those in blue are downregulated in the fish at the late stage of the disease when compared to healthy fish. A quantile color scale was used to show differential expression.

**Figure 4 microorganisms-08-01267-f004:**
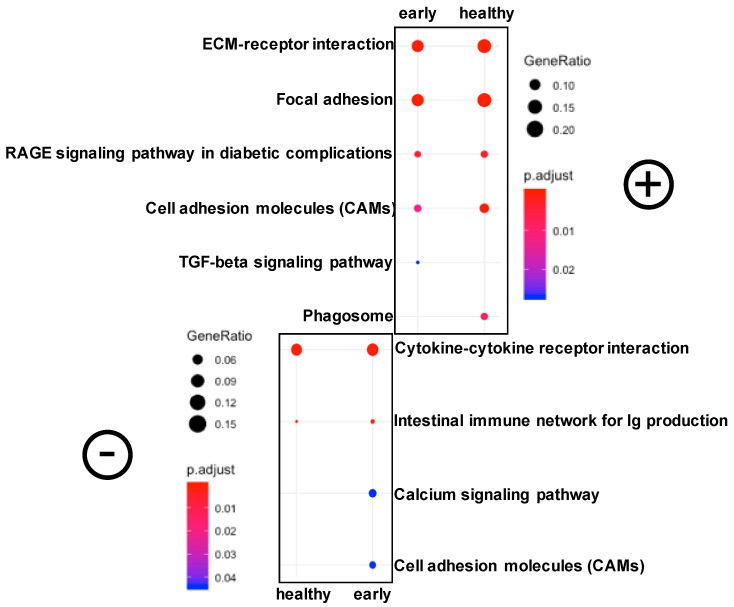
A representation of significantly differentially expressed pathways in the skin of fish at the late stage of the disease compared to both healthy and early conditions using clusterProfiler. Symbols were used to designate upregulated (+) and downregulated (–) pathways.

**Figure 5 microorganisms-08-01267-f005:**
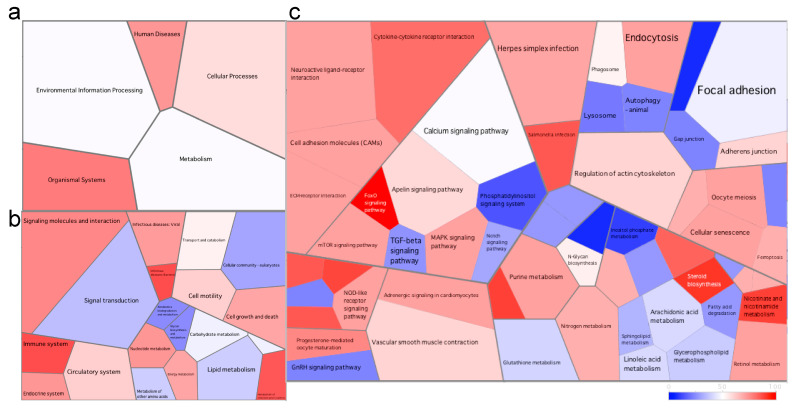
Voronoi tessellation diagrams representing differentially expressed pathways at (**a**) level 1, (**b**) level 2 and (**c**) level 3 of the KEGG database in the skin of the fish at the early stage of the disease. Each polygon represents an ontology term, with their size corresponding to the numbers of genes involved in the associated pathway. To further explore the pathways differentially expressed between health states, we imported all DEGs (and associated logs values) to generate the Voronoi tessellations. Pathways colored in red are upregulated, and those in blue are downregulated in the fish at the early stage of the disease when compared to the healthy fish. A quantile color scale was used to show differential expression.

**Table 1 microorganisms-08-01267-t001:** Table representing the number of differentially expressed genes (DEGs) in the gut and skin of fish exhibiting different health states (H = healthy, E = early stage of enteritis and L = late stage of enteritis).

Sample Type	Gene Expression	HvsE	HvsL	EvsL
gut	upregulated	15	31	18
	downregulated	52	57	1
	total	67	88	19
skin	upregulated	54	481	552
	downregulated	130	986	1516
	total	184	1467	2068

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
