# Peer review of "Investigating Both Mucosal Immunity and Microbiota in Response to Gut Enteritis in Yellowtail Kingfish"

_microorganisms, 2020, doi:10.3390/microorganisms8091267_

Round 1

Reviewer 1 Report

The study of Legrand et al. investigated bacterial shifts across skin and gut mucosal surfaces in yellowtail kingfish showing early and late stage symptoms of intestinal disease and documented the expression of host genes to get a better understanding of the underlying mechanisms contributing to the emergence of an inflammatory disease. As such, this is a very interesting work.

I have few major and minor comments that must be addressed to improve the manuscript in order to be accepted for publication in Microorganisms:

L68: Need an entire paragraph stating the knowledge on microbiota-host immune system interaction at the health/disease border. For instance look at:

McDermott AJ, Huffnagle GB. The microbiome and regulation of mucosal immunity. Immunology. 2014;142(1):24-31. doi:10.1111/imm.12231.

Shi N, Li N, Duan X, Niu H. Interaction between the gut microbiome and mucosal immune system. Mil Med Res. 2017;4:14. Published 2017 Apr 27. doi:10.1186/s40779-017-0122-9.

L199: General comment on Shannon’s diversity index: This is a popular alpha-diversity index because it estimates both richness and evenness in a single equation. However, since its value is dependent of both those parameters, there is theoretically an infinite number of richness / evenness value’ combinations translating into the same index score. Then discuss the result on fig. 1 d: healthy and late are not significantly different, but it does not mean that microbiota composition is similar. Pielou index will be more interesting to compare with raw alpha diversity index such as Chao1, for instance.

L185: General comment on Bray-Curtis distance: this is a metric distance that considers all OTUs to be equally related to each other: it does not take into account relative kinship information between ASVs in the same community, which makes little biological meaning. Therefore, I strongly recommend using the UniFrac distance (both weighted and unweighted, or gunifrac only), which takes into account information on the relative kinship of community members by incorporating the phylogenetic (evolutionary) distances measured between ASVs of the same community. It will provide a better picture in terms of taxonomic differentiation between experimental groups as there is a certain relationship between functional repertories and taxonomy in bacterial communities (Webb et al., 2002; Martiny et al., 2006, 2013; Ward et al., 2006; Gupta and Lorenzini, 2007; Allison and Martiny, 2008; Philippot et al., 2010; Gravel et al., 2011).

L202 : « at the late [stage] of the disease ».

L229: Figure 2: D=diseased cage and H=healthy cage: keep in mind that cages are separated by almost 7 km apart. It is then expected that water community will be differentiated regardless of health status. Replicates are lacking to evaluate geographical and disease status influences on water microbiota composition.

249: There are very few ASVs differentially abundant between healthy and diseased gut, whereas much more ASVs are differentially abundant in skin between healthy and diseased individuals. The fact that the skin microbiota is much more sensitive to the environment than the gut must be discussed (See Sylvain et al. 2019 and 2020 for instance). According to the sampling strategy, environment and disease status are nested: healthy individuals are about 7 km away from diseased individuals, in addition, the water community is different between these two locations. Therefore, the respective effects of the environment and the disease status must be rigorously discussed. For instance, you can argue that many more ASVs were found differentially abundant between early stage of the disease and healthy and late stage of the disease (L252). This argument will be strengthened by making the link with differentially expressed genes associated to immune response = underline those that are known to impact microbiota taxonomic composition.

L455: “Surprisingly, gene expression within the gut did not widely differ between health condition.” Mention that it paralleled the few changes observed in gut microbiota composition. This statement is very interesting, but the link between immune gene expression and microbiota composition must be thoroughly addressed in the discussion and introduction: this is the most interesting result of the study.

To this respect, I strongly recommend to conduct an analysis of correlation between gene expression (RNA-Seq) and AVS’s abundance. This could be achieved by using DSeq2 or by using Cytoscape. Host transcripts can be calculated in reads per kilobase million (RPKM), and ASVs expressed as relative frequency.

Author Response

All comments from the reviewer have been addressed and the manuscript has been updated accordingly except for the addition of the latin name of grass carp which was already stated earlier in the manuscript (L62).

Reviewer 3 Report

The author performed the changes of microbiota across the skin and intestine of diseased yellowtail kingfish by 16s rDNA and RNA-seq approaches. The results showed the intestine of diseased fish was abundant with uncultured Mycoplasmataceae. From data of RNA-seq, there were only some genes significantly differentially expressed in the gut. In contrast, gene expression in the skin differed widely between health states, in particular in the fish at the late stage of the disease. However, this manuscript lack in-depth immunological studies to explore how gut microbiota regulate the host immunity. Thus, the author should add more information, for example: which immunoglobulins play a key role in gut and skin mucosal immunity after infected? Have checked the change of gut mucosa or skin mucosa after infection by H&E staining? Any relationship between the microbiota and immune responses? Moreover, the author should check properly the expressed pathway of intestinal immune network for IgA production in figure 4 as many studies showed teleost do not have immunoglobulin A (Sara Mashoof et al.,2016).

Minor:

  1. The authors may use “gut” or “intestine” throughout the whole manuscript.
  2. The authors mentioned Alpha diversity in lan 151, however, Chao diversity did not appear in this manuscript, it should be checked.

Author Response

- The authors may use “gut” or “intestine” throughout the whole manuscript.

The word intestine was replaced by gut throughout the manuscript.

- The authors mentioned Alpha diversity in lane 151, however, Chao diversity did not appear in this manuscript, it should be checked.

We thank the reviewer for this comment. The manuscript was updated and now include Chao1 diversity in supplementary materials.

- Thus, the author should add more information, for example: which immunoglobulins play a key role in gut and skin mucosal immunity after infected? 

We kindly thank the reviewer for this comment. However, due to the relatively poor annotation of the Seriola dumerili genome used as a reference for this study, such information cannot be provided in the manuscript. In Table S10, we can see some genes related to immunoglobulin differentially expressed (e.g. immunoglobulin lambda-1 light chain-like, immunoglobulin kappa variable 4-1-like). However, it is hard to interpret these genes and therefore we do not further discuss the role of immunoglobulin due to the limited annotation available for this genome.

- Have checked the change of gut mucosa or skin mucosa after infection by H&E staining?

We did not perform any histology of the gut mucosa. However, some histology results of the skin microbiota were reported in our previous publication (Legrand et al. 2018). However, none of these results were relevant for this paper so we did not discuss this further in this manuscript.

- Any relationship between the microbiota and immune responses?

We completely agree with the comment from the reviewer regarding the analysis of correlation between ASVs abundance and gene expression to further explore specific microbe-gene interactions and potentially discovering key genes regulating the host microbiota. Such analysis was performed prior submission of this manuscript using the RcmdrMisc package in R to compute Spearman correlations between these 2 datasets. Unfortunately, due to the relatively low number of replicates for gene expression analysis (3 per treatment), no correlations between genes and ASVs were found from the analysis, as stated in L513-516. We agree that such analysis would be extremely useful to uncover potential host gene-microbe interactions and better understand the influence of both the host immune system and the microbiota during disease. However, such analysis is unfortunately not feasible with this current dataset. Instead, this study highlights the sensitivity of skin (both the host gene expression and the microbiota composition) in response to a gut disease in a farming context. In addition, this study identified potential biomarkers responsible for the loss of microbial diversity in the skin of fish at the early stage of the disease (L447-450, L504-513).

- Moreover, the author should check properly the expressed pathway of intestinal immune network for IgA production in figure 4 as many studies showed teleost do not have immunoglobulin A (Sara Mashoof et al.,2016)

We thank the reviewer for this comment and agree. As such, figure 4 was updated accordingly.